# Despite Vaccination: A Real-Life Experience of Severe and Life-Threatening COVID-19 in Vaccinated and Unvaccinated Patients

**DOI:** 10.3390/vaccines10091540

**Published:** 2022-09-16

**Authors:** Marta Colaneri, Erika Asperges, Matteo Calia, Paolo Sacchi, Marco Rettani, Sara Cutti, Giuseppe Albi, Raffaele Bruno

**Affiliations:** 1Division of Infectious Diseases, Fondazione IRCCS Policlinico San Matteo, 27100 Pavia, Italy; 2Department of Medical, Surgical, Diagnostic and Paediatric Sciences, University of Pavia, 27100 Pavia, Italy; 3Medical Direction, Fondazione IRCCS Policlinico San Matteo, 27100 Pavia, Italy; 4Department of Electrical, Computer and Biomedical Engineering, University of Pavia, 27100 Pavia, Italy

**Keywords:** COVID-19, anti-SARS-CoV-2 vaccinated patients, intra-hospital mortality related to COVID-19, coinfections

## Abstract

Some vaccinated individuals still develop severe COVID-19, and the underlying causes are not entirely understood. We aimed at identifying demographic, clinical, and coinfection characteristics of vaccinated patients who were hospitalized. We also hypothesized that coinfections might play a role in disease severity and mortality. We retrospectively collected data from our COVID-19 registry for whom vaccination data were available. Patients were split into groups based on the number of administered doses (zero, one, two, or three). Data were assessed with Chi-square and Kruskal–Wallis tests and multiple logistic regression analysis. We collected data from 1686 patients and found that intra-hospital mortality was not associated to the vaccination status (e.g., *p* = 0.2 with three doses), while older age, sepsis, and non-viral pneumonia were (*p* < 0.001). Unvaccinated patients needed mechanical ventilation more often (8.5%) than vaccinated patients, in whom the probability of mechanical ventilation decreased with increasing doses (8.7%, 2.8%, 0%). We did not find more coinfections in vaccinated people. We concluded that there is a lack of real-life data to adequately characterize the pathophysiology and risk factors of patients who develop severe COVID-19, but coinfections do not appear to play a role in disease severity.

## 1. Introduction

In addition to the clinical trials of mRNA and adenoviral vector vaccines against SARS-CoV-2 [1,2,3], plenty of real-life data on their high efficacy against COVID-19-related hospitalizations and deaths are currently available [4]. Many Italian reports confirmed these data in the local population, even in the early phase of the vaccination campaign [5,6], and currently, as of June 2022, roughly 48.6 million individuals have received two doses of the COVID-19 vaccine, namely 90% of the total population over 12 years of age [7].

However, some individuals, even if fully vaccinated, develop severe COVID-19, requiring hospital admission. Although certain pre-existing medical conditions, which generally affect the immune response to vaccination [8], might account for the occurrence of post-vaccination severe disease, the underlying causes of such undesirable events are not entirely explored.

On this line of thought, Hippisley and colleagues recently developed a novel clinical risk prediction model to estimate the absolute risk of COVID-19-related death and hospitalization in vaccinated individuals [9], but real-life data are still limited.

Besides vaccination, there are investigations on the impact of additional conditions, such as coinfections, on intra-hospital mortality and the increased severity of the disease [10,11]. In fact, coinfections can be expected to affect the disease course and, possibly, the mortality risk [12].

With this in mind, we aimed at identifying the demographic, clinical, laboratory, and coinfection characteristics of patients who, despite being fully or partially vaccinated with mRNA and adenoviral vector vaccines against SARS-CoV-2, were hospitalized. Moreover, we wanted to investigate these patients’ risk factors for ICU admission and intra-hospital mortality. Finally, we focused on bacterial and viral coinfections detected during hospitalization to better explore a possible association between the coinfection variable and the aforementioned outcomes (ICU admission and intra-hospital mortality).

## 2. Materials and Methods

### 2.1. Study Setting

The SMAtteo COVID-19 REgistry (SMACORE) prospectively collects all the data of COVID-19 patients referred to the IRCCS Policlinico San Matteo Hospital of Pavia, Italy from February 2020, including demography, symptoms at admission and comorbidities, laboratory tests, treatment, and outcomes (admission to the ICU, death or discharge). Ethics approval for observational research using SMACORE data were obtained from the local ethics committee (protocol number 20200046877). In this study, we report a retrospective analysis of SMACORE patients for whom vaccination data were available. The study period runs from January 2021, when the vaccination campaign started in Italy, to March 2022.

The population was split in not vaccinated and vaccinated patients. Vaccinated patients were further classified into 3 groups based on the vaccination type and number of administered doses (1 dose, 2 doses or monodose vaccine, 3 doses).

We only excluded pediatric patients under 5 years of age, given that no vaccine is approved for this age group.

### 2.2. Outcomes

The primary and secondary outcomes of this study were to assess whether the vaccination status has an impact on intra-hospital mortality and ICU admission. First, we compared the characteristics of vaccinated and unvaccinated hospitalized patients. Then, we evaluated the role of bacterial and viral coinfections in the intra-hospital mortality of these patients.

### 2.3. Statistical Analysis

Frequencies were computed for categorical variables, whereas medians and interquartile ranges were employed to describe continuous variables.

To analyze the differences in the demographic, clinical, laboratory, and coinfection variables among the groups, Chi-square tests with Bonferroni-adjusted post-hoc comparisons were employed for categorical variables and Kruskal–Wallis tests with Dunn–Sidak post hoc comparisons for continuous variables.

Multiple logistic regression analyses were performed: the first to assess the association between intra-hospital mortality (independent variable) and vaccination status (dependent variable), including gender, age, number of comorbidities, bacterial or viral coinfection, time elapsed between last vaccination, and hospital admission as confounders. The second regression analysis was performed to assess the association between ICU admission and vaccination status, including the same confounders. The choice of confounders was guided by available data and clinical experience.

The results are expressed in odds ratio (OR) and 95% CI. The significance threshold was set at 0.05. Descriptive analysis was performed with Scikit-learn (version 1.0.2) and SciPy (version 1.7.3) Python’s packages and logistic regression analysis was performed using R (version 4.1.2).

## 3. Results

Overall, we identified 1686 patients, whose characteristics by vaccine status are reported in Table 1. Most patients (1103, 65.4%) did not receive any COVID-19 vaccination, while 92 (5.5%) received only one dose, 321 (19%) received two doses or a single dose of a vaccine approved for a single administration, and 170 (10.1%) received a full three-dose vaccination course. Notably, mRNA-based vaccines were the most frequently used (78.2–88.8%).

Median age was lower in the unvaccinated group than in those who received one, two/monodose, and three doses, with a slight predominance of males in all the analyzed groups.

Regarding comorbidities, the most common was hypertension, followed by diabetes and chronic kidney disease (CKD). CKD, cancer, dementia, and cardiovascular disease were more frequent in those who received three doses of vaccine, compared to not vaccinated patients.

One hundred and forty-four unvaccinated patients (13.1%) died, compared to 17/92 (18.5%, *p* = 0.9), 63/321 (19.6%, *p* = 0.03), and 32/170 (18.8%, *p* = 0.3) patients who received, respectively, one, two/monodose, and three vaccine doses.

Ventilation support need was generally reduced at increasing vaccine doses. Continuous positive airways pressure (CPAP) was used in 35.3% of unvaccinated patients and in just 14.7% of the three-dose vaccinated subpopulation (*p* < 0.001). More importantly, endotracheal intubation (ETI) was never applied in patients who received three vaccine doses.

Similarly, intensive care (ICU) admission rate was lower in the fully vaccinated group (8.2%), with a significantly shorter length of stay (LOS) in the ICU, when compared to those not vaccinated.

Regarding the primary outcome, being vaccinated with two doses/monodose or three doses did not have any impact on the intra-hospital mortality. Age, bacterial sepsis, and non-viral pneumonia were found to be confounders, which resulted in an increased risk of death in this analysis. On the other hand, patients vaccinated with three doses had a lower probability of being admitted to the ICU but again having sepsis, a UTI, or a non-viral pneumonia significantly increased this risk (Table 2).

Data on coinfections are presented in Table 3. The most frequent infectious complications occurring during admission were non-viral pneumonia and UTIs, which occurred homogeneously in the different vaccinal status classes.

## 4. Discussion

According to our results, hospitalized unvaccinated patients were younger and had fewer comorbidities compared to those who were vaccinated with any vaccine dose. We also found that intra-hospital mortality was not associated to the vaccination status, while older age, sepsis, and non-viral pneumonia were associated with a poorer outcome. However, the disease was significantly more severe in unvaccinated patients, who needed continuous positive airway pressure (CPAP) and ETI more often than vaccinated patients. Moreover, the probability of CPAP or ETI was inversely proportional to the doses received, with lower probability in patients who received three doses.

The characteristics of COVID-19 hospitalized patients who did and did not receive any vaccination against SARS-CoV-2 have been sporadically described [13,14] and are substantially in line with our results, showing that vaccinated hospitalized individuals with COVID-19 were older and with a significantly higher burden of comorbidities. Older age and pre-existing comorbidities are well-known major predictors of COVID-19 severity and intra-hospital related mortality [15,16], and we show that, in these circumstances, vaccination is not able to fully prevent negative outcomes, so much so that we documented more deaths in the vaccinated, but older, groups than in younger unvaccinated people. Despite this, and in line with other centers, we confirm that admission for COVID-19 is more likely among unvaccinated individuals [17] who are also more likely to need ICU admission, despite being younger and with fewer major concomitant diseases. This finding is in line with other ICUs’ reports, which claimed a significantly increased proportion of unvaccinated patients [18,19]. Their better health status, thus, clearly cannot outweigh the lack of the immune protection provided by vaccines.

Concordantly, patients with multiple comorbidities, which we observed to be prevalent in the fully vaccinated group, were less likely to end up in the ICU. Although such a result might be justified by the natural inclination to give precedence to younger and less compromised patients [20], it demonstrates the efficacy of vaccines in preventing severe and critical disease, as well as confirming the deadly potential of SARS-CoV-2 even in younger and previously healthy patients, if not adequately protected by vaccination.

The impact of coinfections in COVID-19 outcomes have already been reported [21,22,23,24]. Our results suggest that secondary bacterial infections, in particular, are a significant, but potentially treatable, contributing factor to disease severity and mortality. For this reason, we expected to find a higher prevalence of coinfections in vaccinated people, since this could explain why, despite vaccination, they required admission or even ICU. This was not so, thus, we concluded that when severe COVID-19 occurs in vaccinated people, the worse course of disease is actually due to the lack of vaccine response. The severity of co-occurring infections, however, should still prompt clinicians to design protocols for early diagnosis and treatment [25], especially because the immune dysregulation and functional exhaustion of COVID-19 antiviral lymphocytes [26,27] might also lead to an increased susceptibility to co-infections. On this note, the only coinfections that showed a different prevalence in the four groups were EBV and CMV, which are, in fact, controlled by the lymphocytes, which get depleted during SARS-CoV-2 infection. EBV replication, in particular, seems to be much more common in unvaccinated patients.

Our study is limited by its retrospective and single-center nature. The ample cohort and the inclusion of a large selection of coinfections, however, limits the impact of confounding factors and outliers.

In summary, while the efficacy of anti-SARS-CoV-2 vaccinations can hardly be doubted, there is a lack of real-life data to adequately characterize the pathophysiology and risk factors of patients who, despite being fully or partially vaccinated, are burdened with severe COVID-19. Recognizing these individuals’ risk factors for an adverse outcome might assist clinicians in personalizing treatments and referrals to ICUs.

## Figures and Tables

**Table 1 vaccines-10-01540-t001:** General and clinical characteristics of 1666 patients.

	Not Vaccinated (n = 1103 − 65.4%)	Vaccinated with One Dose (n = 92 − 5.5%)	Vaccinated with Two Doses or Monodose (n = 321 − 19%)	Vaccinated with Three Doses (n = 170 − 10.1%)	*p*-Value
** Demographic and Clinical Data**					
Age (years)	63 (26)	71 (24.5)	73 (24)	75 (19)	<<0.001
Sex					0.63
M	613 (55.6%)	50 (54.3%)	191 (59.5%)	95 (55.9%)	
F	490 (44.4%)	42 (45.7%)	130 (40.4%)	75 (44.1%)	
Vaccine type	-				
mRNA		79 (85.9%)	251 (78.2%)	151 (88.8%)	
		13 (14.1%)	65 (20.2%)	-	
Adenovirus					
mixed			5 (1.6%)	19 (11.2%)	
Comorbidities					
Diabetes	176 (15.9%)	11 (11.9%)	44 (13.7%)	18 (10.6%)	0.22
COPD	58 (5.2%)	12 (13%)	25 (7.8%)	16 (9.4%)	0.006
Hypertension	317 (28.7%)	22 (23.9%)	93 (28.9%)	37 (21.7%)	0.21
CHD	106 (9.6%)	6 (6.5%)	39 (12.1%)	27 (15.9%)	0.03
CLD	31(2.8%)	3 (3.3%)	17 (5.3%)	6 (3.5%)	0.19
CKD	90 (8.2%)	4 (4.3%)	52 (16.2%)	34 (20%)	<<0.001
Cancer	63 (5.7%)	12 (13%)	34 (10.6%)	35 (20.6%)	<<0.001
Dementia	28 (2.5%)	4 (4.3%)	16 (4.9%)	13 (7.6%)	0.004
CVD	65 (5.9%)	5 (5.4%)	51 (15.9%)	30 (17.6%)	<<0.001
Time (days) vaccination-admission (days)	-	21.31 (48.10)	174.16 (92.81)	45.62 (56.73)	<<0.001
** Laboratory Data**					
Lymphocytes, 103	0.6 (0.6)	0.6 (0.6)	0.7 (0.7)	0.6 (0.66)	0.02
Platelets, 103	178 (105)	156.5 (89.75)	165 (103)	161.5 (106.75)	<<0.001
Neutrophils, 103	85.87 (13.12)	85.45 (14.32)	84 (16.2)	87.29 (13.86)	0.003
Creatinine,	0.88 (0.43)	0.93 (0.52)	1.08 (0.66)	1.16 (1.01	<<0.001
CRP,	9.97 (12.61)	10.92 (12.31)	9.65 (13.89)	7.33 (14.52)	0.46
PCTI,	0.17 (0.82)	0.19 (0.87)	0.32 (1.71)	0.68 (3.2)	<<0.001
DD,	1822 (3948)	1810 (3392)	1949 (5657)	2099 (3934)	0.52
Glucose,	112 (52)	108.5 (38.25)	113 (52.2)	121 (63)	0.1
TNI,	14 (58.5)	39 (81.25)	28.5 (176.25)	76.5 (519.25)	<<0.001
** Clinical Outcomes**					
Death	144 (13.1%)	17 (18.5%)	63 (19.6%)	32 (18.8%)	0.01
Oxygen required					
CPAP	392 (35.3%)	27 (29.3%)	66 (20.5%)	25 (14.7%)	<<0.001
ETI	94 (8.5%)	8 (8.7%)	9 (2.8%)	-	<0.001
ILOS (days)	12.31 (14.13)	12.41 (14.54)	10.09 (14.09)	11.14 (13.93)	0.02
ICU admitted					
yes	267 (24.2%)	15 (16.3%)	22 (6.8%)	14 (8.2%)	<<0.001
ICU-LOS (days)	12.98 (21.22)	13.28 (23.54)	12.52 (31.97)	6.95 (5.55)	<<0.001

COPD, chronic obstructive pulmonary disease; CHD, coronary heart disease; CLD, chronic liver disease; CKD, chronic kidney disease; CVD, cerebrovascular disease; CPAP, continuous positive airways pressure; ETI, endotracheal intubation; ICU, intensive care unit; LOS, length of stay; DD: D-dimer; PCT: procalcitonin; CRP: C-reactive protein; TNI: troponin.

**Table 2 vaccines-10-01540-t002:** Multivariable logistic regression analysis of factors associated to death and ICU admission.

	OR	95% CI	*p*-Value
Analysis on intra-hospital mortality			
Vaccinated with one dose	0.91	0.46–1.72	0.795
Vaccinated with two doses or monodose	0.63	0.29–1.30	0.227
Vaccinated with three doses	0.73	0.43–1.22	0.243
Age	1.09	1.07–1.10	<0.001
Sex (male)	1.00	0.72–1.38	0.982
Number of comorbidities	1.17	0.99–1.39	0.053
Time between last vaccinations and admission (days)	1.00	0.99–1.00	0.148
BSIs	3.22	1.99–5.23	<0.001
UTIs	0.79	0.52–1.17	0.26
Clostridioides colitis	0.99	0.24–3.50	0.995
Non-viral pneumonia	3.82	2.60–5.62	<0.001
Analysis on ICU admission			
Vaccinated with one dose	0.48	0.21–1.01	0.066
Vaccinated with two doses or monodose	0.27	0.09–0.68	0.008
Vaccinated with three doses	0.31	0.15–0.61	0.001
Age	1.00	0.99–1.01	0.620
Sex (male)	1.53	1.12–2.11	0.008
Number of comorbidities	0.76	0.62–0.92	0.006
Time between last vaccinations and admission (days)	0.99	0.99–1.00	0.128
Sepsis	5.56	3.49–8.89	<0.001
UTIs	2.65	1.77–3.96	<0.001
Clostridioides colitis	0.52	0.09–2.12	0.401
Non-viral pneumonia	6.00	4.23–8.51	<0.001

BSIs: blood stream infections; ICU: intensive care unit; UTIs: urinary tract infections.

**Table 3 vaccines-10-01540-t003:** Data about coinfections by vaccination status.

	Not Vaccinated n = 1103 (65.4%)	Vaccinated with One Dose n = 92 (5.5%)	Vaccinated with Two Doses or Monodose n = 321 (19%)	Vaccinated with Three Doses n = 170 (10.1%)	*p*-Value
BSIs	121 (10.9%)	14 (15.2%)	31 (9.6%)	10 (5.9%)	0.09
MDR colonization	86 (7.8%)	7 (7.6%)	16 (4.9%)	8 (4.7%)	0.21
NVP	240 (21.7%)	20 (21.7%)	64 (19.9%)	37 (21.7%)	0.92
UTI	178 (16.1%)	17 (18.5%)	56 (17.4%)	25 (14.7%)	0.81
Clostridioides colitis	8 (0.7%)	1 (1.1%)	2 (0.6%)	3 (1.7%)	0.53
CMV	29 (2.6%)	3 (3.3%)	2 (0.6%)	10 (5.9%)	0.006
EBV	134 (12.1%)	6 (6.5%)	14 (4.4%)	10 (5.9%)	<<0.001
HIV	3 (0.3%)	-	1 (0.3%)	-	0.86
Other respiratory viruses	15 (1.3%)	1 (1.1%)	1 (0.3%)	1 (0.6%)	0.39

MDR, multi drug resistant; NVP, non-viral pneumonia; UTI, urinary tract infection; CMV, cytomegalovirus; EBV: Epstein–Barr virus; HIV: human immunodeficiency virus.

## Data Availability

The published article includes all datasets generated or analyzed during the study. Data sharing is available from the corresponding author upon reasonable request.

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
