# Peer review of "Despite Vaccination: A Real-Life Experience of Severe and Life-Threatening COVID-19 in Vaccinated and Unvaccinated Patients"

_vaccines, 2022, doi:10.3390/vaccines10091540_

Round 1
Reviewer 1 Report
This is quite exciting work. The paper is written quite well and thoroughly. The message is clear.
Author Response
We thank the reviewer for the kind comment.
Reviewer 2 Report
-Abstract and introduction: The authors described that "we aimed at identifying the demographic, clinical, laboratory, and coinfection characteristics of patients who, despite being fully or partially vaccinated with mRNA and adenoviral vector vaccines against SARS-CoV-2, were hospitalized." But in abstract the aim is incomplete. In addition, no numbers of results are provided in the abstract. What is hypothesis of study?
-Methods: Methods are unclear and lack informations. How the data were collected? Disease? Biochemical data? ICU data?
-Results and Discussion: No new informations were provided. Moreover, discussion is vague "The characteristics of COVID-19 hospitalized patients who did and did not receive any vaccination against SARS-CoV-2 have been sporadically described".
-What are positive and negative points of study?
Author Response
We thank the reviewer for the comments.
- We agreed on the deficiencies the reviewer pointed out and have improved the abstract completing the aim and adding the hypothesis and numbers in the results.
- Methods: in the first paragraph we explain how the data come from the hospital registry of Covid19 patients, were all the data including biochemical analysis, comorbidities and treatments (including ventilation and/or oxygen therapy) are collected. We feel that further expanding on the methods of data collection of a registry does not provide additional information.
- Discussion: we thank you the reviewer for pointing out that we did not include strengths and weaknesses. This has been amended. We also added a further comment on CMV and EBV coinfections to expand the discussion. However, we respectfully feel that we did provide adequate information and insights on our results. We are not sure what the reviewer means us to do with the sentence in the quotation marks.
Reviewer 3 Report
An interesting and well-written manuscript investigating why some vaccinated patients (in comparison to vaccinated) have the worst outcomes (i.e. death or ICU admission). The manuscript is in general well designed and written. Just a few comments that may improve it.
Line 91 “Data analysis was performed mainly using R” why mainly? please describe all tools used for statistics.
Lines 108-9 “One hundred forty-four not vaccinated patients (13.1%) died, compared to 17/92 108 (18.5%), 63/321 (19.6%, p=0.03) and 32/170 (18.8%) patients who received respectively 1, 109 2/monodose and 3 vaccine doses.”. the p value of 0.03 is probably when compared with the non vaccinated population that died.!! However please clarify. Moreover add p-values for the other pairs: 1-dose vs. non-vaccination, and 3 doses vs non-vaccination. It is strange that the percentage of vaccinated patients that died is higher (they probably come from high risk groups), in any case, vaccination doses is not related to death in the multivariable analysis.
Table 2, please clarify all acronyms (what is BSI?), moreover the horizontal lines do not always separate variables of the same group (for example vaccine doses or demographics).
Table 3, an additional column with statistical significance for each line would be of interest, to show possible driving factors for vaccination.
Section statistical analysis, please provide more details for the logistic regression process (forward? backward? what was the p-value to let a variable stay in the model?)
Author Response
Thank your for your feedback.
Line 91: We added details of all the tools we used for the analyses.
Line 108: we added the request p values and clarified the comparison. In the second paragraph of the discussion we now expanded on how we attribute the increased percentage of vaccinated deaths to their much older age (vaccinated people were significantly younger).
Table 2: corrected.
Table 3: column added as requested.
Statistical analysis: the choice of cofounders was guided by clinical experience and practicality. An explanation was added in the text.
Round 2
Reviewer 2 Report
Low quality of manuscript
Author Response
We thank the reviewer for their time. Before making any changes, however, we would like to ask if they can be more specific in how the manuscript can be improved since their comment does not say whether they detected problems in the methods, the conclusions or any other sections. We are amenable to providing more data or analyses if this is what the reviewer meant.